# CLUSTERED REINFORCEMENT LEARNING

## ABSTRACT

Exploration strategy design is one of the challenging problems in reinforcement learning (RL), especially when the environment contains a large state space or sparse rewards. During exploration, the agent tries to discover novel areas or high reward (quality) areas. In most existing methods, the novelty and quality in the neighboring area of the current state are not well utilized to guide the exploration of the agent. To tackle this problem, we propose a novel RL framework, called clustered reinforcement learning (CRL), for efficient exploration in RL. CRL adopts clustering to divide the collected states into several clusters, based on which a bonus reward reflecting both novelty and quality in the neighboring area (cluster) of the current state is given to the agent. Experiments on several continuous control tasks and several Atari-2600 games show that CRL can outperform other state-of-the-art methods to achieve the best performance in most cases.

## 1 INTRODUCTION

Reinforcement learning (RL) (Sutton & Barto, 1998) studies how an agent can maximize its cumulative reward in an unknown environment, by learning through exploration and exploitation. A key challenge in RL is to balance the relationship between exploration and exploitation. If the agent explores novel states excessively, it might never find rewards to guide the learning direction. Otherwise, if the agent exploits rewards too intensely, it might converge to suboptimal behaviors and have fewer opportunities to discover more rewards from exploration.

Although reinforcement learning, especially deep RL (DRL), has recently attracted much attention and achieved significant performance in a variety of applications, such as game playing (Mnih et al., 2015; Silver et al., 2016) and robot navigation (Zhang et al., 2016), exploration techniques in RL are far from satisfactory in many cases. Exploration strategy design is still one of the challenging problems in RL, especially when the environment contains a large state space or sparse rewards. Hence, it has become a hot research topic to design exploration strategy, and many exploration methods have been proposed in recent years.

Some heuristic methods for exploration, such as $\epsilon$-greedy (Silver et al., 2016; Sutton & Barto, 1998), uniform sampling (Mnih et al., 2015) and i.i.d./correlated Gaussian noise (Lillicrap et al., 2016; Schulman et al., 2015), try to directly obtain more different experiences during exploration. For hard applications or games, these heuristic methods are insufficient enough and the agent needs exploration techniques that can incorporate meaningful information about the environment.

In recent years, some exploration strategies try to discover novel state areas for exploring. The direct way to measure novelty is to count the visited experiences. In (Bellemare et al., 2016; Ostrovski et al., 2017), pseudo-counts are estimated from a density model. Hash-based method (Tang et al., 2017) counts the hash codes of states. There also exist some work using the counts of state-action pairs to design their exploration techniques, such as explicit explore or exploit ($E^3$) (Kearns & Singh, 2002), R-Max Brafman & Tennenholtz (2002), UCRL (Auer & Ortner, 2006), UCAGG (Ortner, 2013). Besides, the state novelty can also be measured by empowerment (Klyubin et al., 2005), the agent's belief of environment dynamics (Houthooft et al., 2016), prediction error of the system dynamics model (Pathak et al., 2017; Stadie et al., 2015), prediction by exemplar model (Fu et al., 2017), and the error of predicting features of states (Burda et al., 2018). All the above methods perform exploration mainly based on the novelty of states without considering the quality of states. Furthermore, there are some methods to estimate the quality of states. Kernel-based reinforcement

learning (Ormoneit & Sen, 2002) uses locally weighted averaging to estimate the quality (value) of states. UCRL (Auer & Ortner, 2006) and UCAGG (Ortner, 2013) compute average rewards for choosing optimistic values. The average reward can be regarded as an estimation of the quality of states to guide the exploring direction, but there are no methods using the quality of states as an exploration technique. Furthermore, in most existing methods, the novelty and quality in the neighboring area of the current state are not well utilized to guide the exploration of the agent.

To tackle this problem, we propose a novel RL framework, called clustered reinforcement learning (CRL), for efficient exploration in RL. The contributions of CRL are briefly outlined as follows:

- CRL adopts clustering to divide the collected states into several clusters. The states from the same cluster have similar features. Hence, the clustered results in CRL provide a possibility to share meaningful information among different states from the same cluster.

- CRL proposes a novel bonus reward, which reflects both novelty and quality in the neighboring area of the current state. Here, the neighboring area is defined by the states which share the same cluster with the current state. This bonus reward can guide the agent to perform efficient exploration, by seamlessly integrating novelty and quality of states.

- Experiments on several continuous control tasks with sparse rewards and several hard exploratory Atari-2600 games (Bellemare et al., 2013) show that CRL can outperform other state-of-the-art methods to achieve the best performance in most cases. In particular, on several games known to be hard for heuristic exploration strategies, CRL achieves significantly improvement over baselines.

## 2 RELATED WORK

Recently, there are some exploration strategies used to discover novel state areas. The direct way to measure the novelty of states is to count the visited experiences, which has been applied in several methods. In the tabular setting and finite Markov decision processes (MDPs), the number of state-action pairs is finite which can be counted directly, such as model-based interval estimation with exploratory bonus (MBIE-EB) (Strehl & Littman, 2008), explicit explore or exploit ($E^3$) (Kearns & Singh, 2002) and R-Max (Brafman & Tennenholtz, 2002). MBIE-EB adds the reciprocal of square root of counts of state-action pairs as the bonus reward to the augmented Bellman equation for exploring less visited ones with theoretical guarantee. $E^3$ determines the action based on the counts of state-actions pairs. If the state has never been visited, the action is chosen randomly and if the state has been visited for some times, the agent takes the action that has been tried the fewest times before. R-Max uses counts of states as a way to check for known states.

In the continuous and high-dimensional space, the number of states is too large to be counted directly (Bellemare et al., 2016; Ostrovski et al., 2017; Tang et al., 2017; Abel et al., 2016). Bellemare et al. (2016) and Ostrovski et al. (2017) use a density model to estimate the state pseudo-count quantity, which is used to design the exploration bonus reward. Tang et al. (2017) counts the number of states by using the hash function to encode states and then it explores by using the reciprocal of visits as a form of reward bonus, which performs well on some hard exploration Atari-2600 games. Abel et al. (2016) records the number of cluster center and action pairs and makes use of it to select an action from the Gibbs distribution. These count-based methods encourage the agent to explore by making use of the novelty of states and do not take quality into consideration.

Furthermore, there are some methods to estimate the quality of states. Average reward, in kernel-based reinforcement learning (Ormoneit & Sen, 2002), UCRL (Auer & Ortner, 2006) and UCAGG (Ortner, 2013), can be regarded as the quality of states. Kernel-based reinforcement learning (Ormoneit & Sen, 2002) is proposed to solve the stability problem of TD-learning by using locally weighted averaging to estimate the value of state. UCRL (Auer & Ortner, 2006) and UCAGG (Ortner, 2013) use average reward to choose optimistic values. Besides, the value of cluster space can also indicate the quality of states. Singh et al. (1994) uses the value of cluster space with Q-learning and TD(0) by soft state aggregation and provides convergence results. But these methods do not use the quality of states to explore more areas.

To the best of our knowledge, the novelty and quality in the neighboring area of the current state have not been well utilized to guide the exploration of the agent in existing methods, especially in the high dimensional state space. This motivates the work of this paper.

## 3 NOTATION

In this paper, we adopt similar notations as those in (Tang et al., 2017). More specifically, we model the RL problem as a finite-horizon discounted Markov decision process (MDP), which can be defined by a tuple $(\mathcal{S}, \mathcal{A}, \mathcal{P}, r, \rho_0, \gamma, T)$. Here, $\mathcal{S} \in \mathbb{R}^d$ denotes the state space, $\mathcal{A} \in \mathbb{R}^m$ denotes the action space, $\mathcal{P} : \mathcal{S} \times \mathcal{A} \times \mathcal{S} \to \mathbb{R}$ denotes a transition probability distribution, $r : \mathcal{S} \times \mathcal{A} \to \mathbb{R}$ denotes a reward function, $\rho_0$ is an initial state distribution, $\gamma \in (0, 1]$ is a discount factor, and $T$ denotes the horizon time. In this paper, we assume $r \geq 0$. For cases with negative rewards, we can transform them to cases without negative rewards. The goal of RL is to maximize $\mathbb{E}_{\pi, \mathcal{P}} \left[ \sum_{t=0}^{T} \gamma^t r\left(s_t, a_t\right) \right]$ which is the total expected discounted reward over a policy $\pi$.

## 4 CLUSTERED REINFORCEMENT LEARNING

This section presents the details of our proposed RL framework, called clustered reinforcement learning (CRL). The key idea of CRL is to adopt clustering to divide the collected states into several clusters, and then design a novel cluster-based bonus reward for exploration.

### 4.1 CLUSTERING

Intuitively, both novelty and quality are useful for exploration strategy design. If the agent only cares about novelty, it might explore intensively in some unexplored areas without any reward. If the agent only cares about quality, it might converge to suboptimal behaviors and have low opportunity to discover unexplored areas with higher rewards. Hence, it is better to integrate both novelty and quality into the same exploration strategy.

We find that clustering can provide the possibility to integrate both novelty and quality together. Intuitively, a cluster of states can be treated as an area. The number of collected states in a cluster reflects the count (novelty) information of that area. The average reward of the collected states in a cluster reflects the quality of that area. Hence, based on the clustered results, we can design an exploration strategy considering both novelty and quality. Furthermore, the states from the same cluster have similar features, and hence the clustered results provide a possibility to share meaningful information among different states from the same cluster. The details of exploration strategy design based on clustering will be left to the following subsection. Here, we only describe the clustering algorithm.

In CRL, we perform clustering on states. Assume the number of clusters is $K$, and we have collected $N$ state-action samples $\{(s_i, a_i, r_i)\}_{i=1}^{N}$ with some policy. We need to cluster the collected states $\{s_i\}_{i=1}^{N}$ into $K$ clusters by using some clustering algorithm $f : \mathcal{S} \to \mathcal{C}$, where $\mathcal{C} = \{C_i\}_{i=1}^{K}$ and $C_i$ is the center of the $i$-th cluster. We can use any clustering algorithm in the CRL framework. Although more sophisticated clustering algorithms might be able to achieve better performance, in this paper we just choose k-means algorithm (Coates & Ng, 2012). K-means is one of the simplest clustering algorithms with wide applications. The detail of k-means is omitted here, and readers can find it in most machine learning textbooks.

### 4.2 CLUSTERING-BASED BONUS REWARD

As stated above, clustering can provide the possibility of integrating both novelty and quality together for exploration. Here, we propose a novel clustering-based bonus reward, based on which many *policy updating algorithms* can be adopted to get an exploration strategy considering both novelty and quality.

Given a state $s_i$, it will be allocated to the nearest cluster by the cluster assignment function $\phi(s_i) = \arg\min_{k} \|s_i - C_k\|$. Here, $1 \leqslant k \leqslant K$ and $\|s_i - C_k\|$ denotes the distance between $s_i$ and the $k$-th

cluster center $C_k$. The sum of rewards in the $k$-th cluster is denoted as $R_k$, which can be computed as follows:

$$R_k = \sum_{i=1}^{N} r_i \mathbb{I}(\phi(s_i) = k), \qquad (1)$$

where $\mathbb{I}(\cdot)$ is an indicator function. $R_k$ is also called *cluster reward* of cluster $k$ in this paper. The number of states in the $k$-th cluster is denoted as $N_k$, which can be computed as follows:

$$N_k = \sum_{i=1}^{N} \mathbb{I}(\phi(s_i) = k). \qquad (2)$$

Intuitively, a larger $N_k$ typically means that the area corresponding to cluster $k$ has more visits (exploration), which implies the novelty of this area is lower. Hence, the bonus reward should be inversely proportional to $N_k$. The average reward of cluster $k$, denoted as $\frac{R_k}{N_k}$, can be used to represent the quality of the corresponding area of cluster $k$. Hence, the bonus reward should be proportional to $\frac{R_k}{N_k}$.

With the above intuition, we propose a clustering-based bonus reward $b : \mathcal{S} \to \mathbb{R}$ to integrate both novelty and quality of the neighboring area of the current state $s$, which is defined as follows:

$$b(s) = \beta \frac{\max(\eta, R_{\phi(s)})}{N_{\phi(s)}}, \qquad (3)$$

where $\beta \in \mathbb{R}^+$ is the bonus coefficient and $\eta \in \mathbb{R}^+$ is the count (novelty) coefficient. Typically, $\eta$ is set to be a small number relative to a true reward [1].

In general, as long as there exist one or two states with positive rewards in cluster $\phi(s)$, $R_{\phi(s)}$ will larger than $\eta$. Hence, if $b(s) = \frac{\beta \eta}{N_{\phi(s)}}$, it is highly possible that all states in cluster $\phi(s)$ have zero reward. Hence, when $R_{\phi(s)} = 0$ which means no rewards have been got for cluster $\phi(s)$, the bonus reward should be determined by the count of the cluster. From Equation (3), a larger $N_{\phi(s)}$ will result in a smaller bonus reward $b(s)$. This will guide the agent to explore novel area corresponding to clusters with less visits (exploration), which is reasonable. For two clusters with the same cluster reward, the cluster with smaller number of states (higher novelty) will be more likely to be explored, which is reasonable. For two clusters with the same number of states, the cluster with higher cluster reward (higher quality) will be more likely to be explored, which is also reasonable.

Hence, the clustering-based bonus reward function defined in Equation (3) is intuitively reasonable, and it can seamlessly integrate both novelty and quality into the same bonus function. Finally, the agent will adopt $\{(s_i, a_i, r_i + b_i)\}_{i=1}^{N}$ to update the policy (perform exploration). Many *policy updating algorithms*, such as trust region policy optimization (TRPO) (Schulman et al., 2015), can be adopted.

Algorithm 1 briefly presents the learning framework of CRL. We can see that CRL is actually a general framework, and we can get different RL variants by taking different clustering algorithms and different policy updating algorithms. Please note that $r_i + b_i$ is only used for training Algorithm 1. But the performance evaluation (test) is measured without $b_i$, which can be directly compared with existing RL methods without extra bonus reward.

## 5 EXPERIMENTS

We use several continuous control tasks and several Atari-2600 games to evaluate CRL and baselines. We want to investigate and answer the following research questions:

- Is the count-based exploration sufficient to encourage the agent to achieve the final goal of tasks?
- Can CRL improve performance significantly across different tasks compared with other methods?

---

[1] In our experiments, the true rewards are either zero or positive integers.

---

**Algorithm 1** Framework of Clustered Reinforcement Learning (CRL)

Initialize the number of clusters $K$, bonus coefficient $\beta$, count coefficient $\eta$
**for** iteration $j = 1, \ldots, J$ **do**
  Collect a set of state-action samples $\{(s_i, a_i, r_i)\}_{i=1}^{N}$ with policy $\pi_j$;
  Cluster the state samples with $f : \mathcal{S} \to \mathcal{C}$, where $\mathcal{C} = \{C_i\}_{i=1}^{K}$ and $f$ is some *clustering algorithm*;
  Compute the cluster assignment for each state $\phi(s_i) = \arg\min_{k} \|s_i - C_k\|, \forall i : 1 \leqslant i \leqslant N, \ k : 1 \leqslant k \leqslant K$;
  Compute sum of rewards $R_k$ using Equation (1) and the number of states $N_k$ using Equation (2), $\forall k : 1 \leqslant k \leqslant K$;
  Compute the bonus $b(s_i)$ using Equation (3);
  Update the policy $\pi_j$ using rewards $\{r_i + b(s_i)\}_{i=1}^{N}$ with some *policy updating algorithm*;
**end for**

---

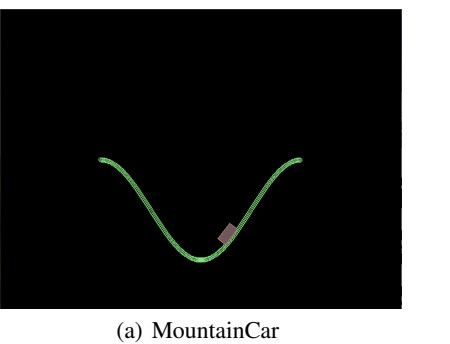

(a) MountainCar

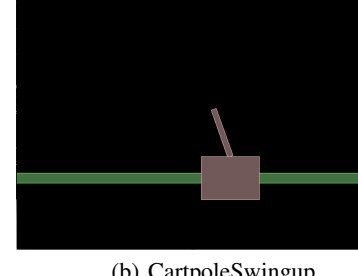

(b) CartpoleSwingup

Figure 1: Snapshots of two *MuJoCo* games.

## 5.1 EXPERIMENTAL SETUP

### 5.1.1 ENVIRONMENTS

*MuJoCo.* The rllab benchmark (Duan et al., 2016) consists of various continuous control tasks to test RL algorithms. We select MountainCar and CartpoleSwingup to compare our methods with other baselines. The experimental setups of MountainCar and CartPoleSwingup using sparse rewards can be found in Houthooft et al. (2016). In MountainCar, $\mathcal{S} \subseteq \mathbb{R}^3, \mathcal{A} \subseteq \mathbb{R}^1$. The agent receives a reward of $+1$ when the car escapes the valley from the right side, otherwise the agent receives a reward of $0$. In CartpoleSwingup, $\mathcal{S} \subseteq \mathbb{R}^4, \mathcal{A} \subseteq \mathbb{R}^1$. The agent receives a reward of $+1$ when the cosine of pole angle is larger than $0.8$, otherwise the agent receives a zero return at other positions. Figure 1 shows one snapshot for each task.

*Arcade Learning Environment.* The Arcade Learning Environment (ALE) (Bellemare et al., 2013) is a commonly used benchmark for RL algorithms because of its high-dimensional state space and wide variety of video games. We select a subset of Atari games[2]: Freeway, Frostbite, Gravitar, Solaris and Venture. Figure 2 shows a snapshot for each game. For example, in Freeway, the agent need to avoid the traffic, cross the road and get the reward. These games are classified into hard exploration category, according to the taxonomy in (Bellemare et al., 2016).

### 5.1.2 BASELINES

CRL is a general framework which can adopt many different *policy updating (optimization) algorithms* to get different variants. In this paper, we only adopt trust region policy optimiza-

---

[2]The Montezuma game evaluated in Tang et al. (2017) is not adopted in this paper for evaluation, because this paper only uses raw pixels which are not enough for learning effective policy on Montezuma game for most methods including CRL and other baselines. We can use advanced feature to learn effective policy, but this is not the focus of this paper.

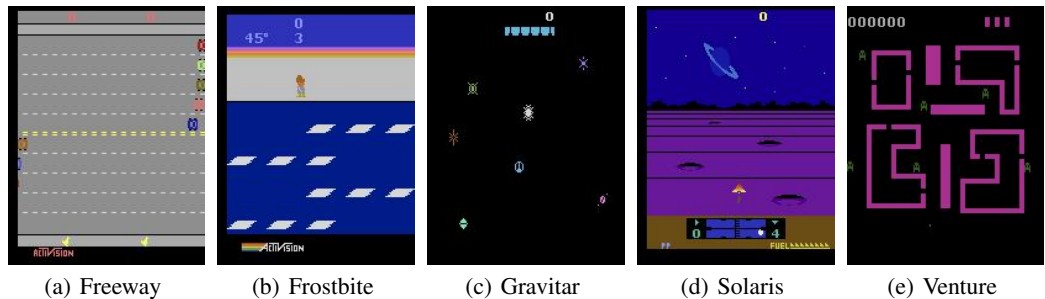

| (a) Freeway | (b) Frostbite | (c) Gravitar | (d) Solaris | (e) Venture |

Figure 2: Snapshots of five hard exploration Atari-2600 games.

tion (TRPO) (Schulman et al., 2015) as the policy updating algorithm for CRL, and leave other variants of CRL for future work. We will denote our method as $CRL_{TRPO}$ in the following content. The baselines for comparison include TRPO and TRPO-Hash (Tang et al., 2017). For continuous control problem, we choose VIME as a baseline.

*TRPO* (Schulman et al., 2015) is a classic policy gradient method, which uses trust region to guarantee stable improvement of policy and can handle both discrete and continuous action space. Furthermore, this method is not too sensitive to hyper-parameters. TRPO adopts a Gaussian control noise as a heuristic exploration strategy.

*TRPO-Hash* (Tang et al., 2017) is a hash-based method, which is a generalization of classic count-based method for high-dimensional and continuous state spaces. The main idea is to use locality-sensitive hashing (LSH) (Andoni & Indyk, 2006) to encode continuous and high-dimensional data into hash codes, like $\{-1, 1\}^h$. Here, $h$ is the length of hash codes. TRPO-Hash has several variants in (Tang et al., 2017). For fair comparison, we choose SimHash (Charikar, 2002) (TRPO-Hash) as the hash function and pixels as inputs for TRPO-Hash in this paper, because our CRL also adopts pixels rather than advanced features as inputs. TRPO-Hash is trained by using the code provided by its authors.

*VIME* (Houthooft et al., 2016) is a curiosity-driven exploration strategy, which seeks out unexplored state-action region by maximizing the information gain of agent's belief of environments. VIME is also trained by using the code provided by its authors. Here, we select VIME to compare with our method in continuous control problem because this method only supports continuous state and action space.

## 5.2 PERFORMANCE ON MUJOCO

Figure 3 shows the results of TRPO, TRPO-Hash, VIME and $CRL_{TRPO}$ in MountainCar and CartpoleSwingup. We can find that our $CRL_{TRPO}$ achieves the best performance on both MountainCar and CartpoleSwingup. In MountainCar, our method is the first one to reach the goal state and master a good policy. Our method outperforms all other methods with a large margin. The goal of TRPO-Hash is to help the agent explore more novel states. But TRPO-Hash might go through all states until reaching the goal state, which is the disadvantage of count (novelty) based exploration. We find that at the end of training, TRPO-Hash fails to achieve the goal that our method and VIME have achieved. The reason why TRPO-Hash fails is that the novelty of states diverts the agent's attention. The worst case is that the agent collects all states until it finds the goal. This disadvantage of count-based methods might become more serious in the high-dimensional state space since it is impossible to go through all states in the high-dimensional state space. Therefore, strategies with only count-based exploration are insufficient.

## 5.3 PERFORMANCE ON ATARI-2600

For the video games of Atari-2600, we compare $CRL_{TRPO}$ with other baselines. The agent is trained for 500 iterations in all experiments with each iteration consisting of 0.4M frames. The agent selects an action every 4 frames, so every iteration consists of 0.1M steps (0.4M frames). The

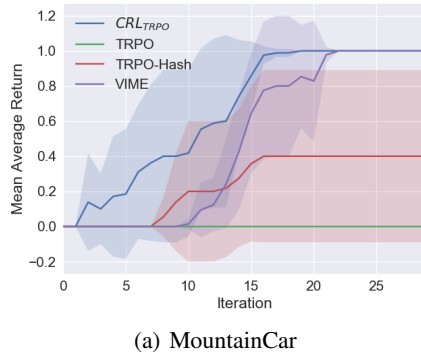

(a) MountainCar

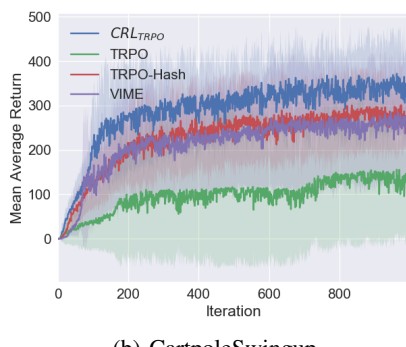

(b) CartpoleSwingup

Figure 3: Mean average return of different algorithms on MountainCar and CartpoleSwingup over 5 random seeds. The solid line represents the mean average return and the shaded area represents one standard deviation.

Table 1: The mean average undiscounted return after training for 50M time steps (200M frames).

|  | Freeway | Frostbite | Gravitar | Solaris | Venture |
|---|---|---|---|---|---|
| TRPO | 17.55 | 1229.66 | 500.33 | 2110.22 | 283.48 |
| TRPO-Hash | 22.29 | 2954.10 | **577.47** | 2619.32 | 299.61 |
| CRL$_{TRPO}$ | **26.68** | **4558.52** | 541.72 | **2976.23** | **723.94** |
| Double-DQN | 33.3 | 1683 | 412 | 3068 | 98.0 |
| Dueling network | 0.0 | 4672 | 588 | 2251 | 497 |
| A3C+ | 27.3 | 507 | 246 | 2175 | 0 |
| pseudo-count | 29.2 | 1450 | - | - | 369 |

last frames of every 4 frames are used for clustering and counting. The performance is evaluated over 5 random seeds. The seeds for evaluation are the same for all methods.

We summarize all results in Table 1. Please note that TRPO and TRPO-Hash are trained with the code provided by the authors of TRPO-Hash. All hyper-parameters are reported in the supplementary material. We also compare our methods to double-DQN (van Hasselt et al., 2016), dueling network (Wang et al., 2016), A3C+ (Bellemare et al., 2016), double DQN with pseudo-count (Bellemare et al., 2016), the results of which are from (Tang et al., 2017). Furthermore, we show the training curves of our methods, TRPO and TRPO-Hash in Figure 4.

CRL$_{TRPO}$ achieves significantly improvement over TRPO and TRPO-Hash on Freeway, Frostbite, Solaris and Venture. Please note that DQN-based methods reuse off-policy experience. Hence, DQN-based methods have better performance than TRPO without any exploration techniques in most cases. But our methods can still outperform DQN-based methods in most cases.

## 6 CONCLUSION

In this paper, we propose a novel RL framework, called clustered reinforcement learning (CRL), for efficient exploration. By using clustering, CRL provides a general framework to adopt both novelty and quality in the neighboring area of the current state for exploration. Experiments on several continuous control tasks and several hard exploration Atari-2600 games show that CRL can outperform other state-of-the-art methods to achieve the best performance in most cases.

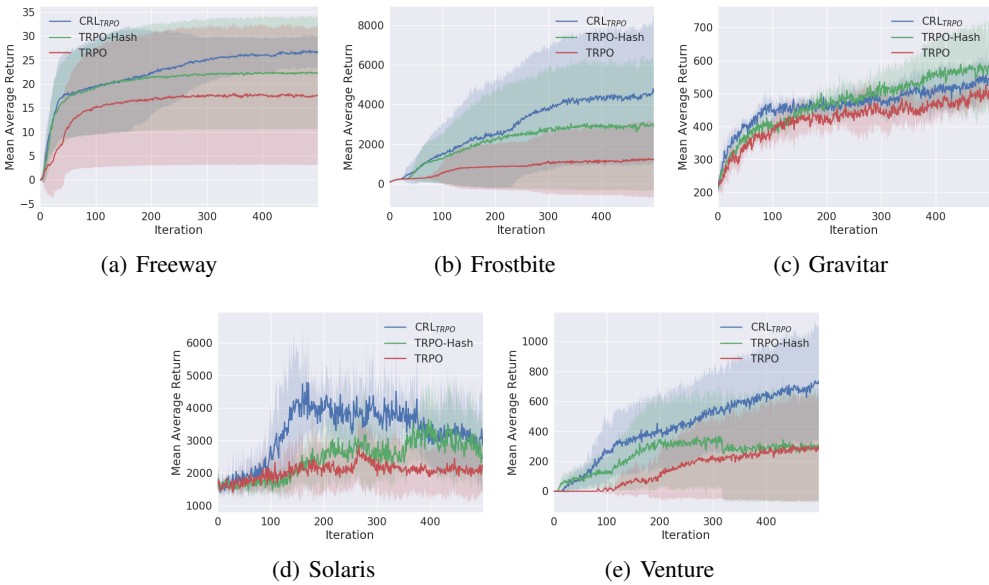

(a) Freeway  (b) Frostbite  (c) Gravitar

(d) Solaris  (e) Venture

Figure 4: Mean average return of different algorithms on Atari-2600 over 5 random seeds. The solid line represents the mean average return and the shaded area represents one standard deviation.

# A APPENDIX

## A.1 HYPER-PARAMETER SETTING IN MuJoCo

In MuJoCo, the hyper-parameter setting of TRPO, TRPO-Hash, CRL$_{TRPO}$ is shown in Table 2. The hyper-parameter setting of VIME can be found in (Houthooft et al., 2016). The performance is evaluated over 5 random seeds. The seeds for evaluation are the same for all methods.

Table 2: Hyper-parameter setting in MuJoCo.

|  | TRPO | TRPO-Hash | CRL$_{TRPO}$ |
|---|---|---|---|
| TRPO batchsize | | 5000 | |
| TRPO stepsize | | 0.01 | |
| Discount factor | | 0.99 | |
| Policy hidden units | | (32, 32) | |
| Baseline function | | Linear | |
| Iteration | | 30 | |
| Max length of path | | 500 | |
| Bonus coefficient | - | 0.01 | 1 |
| Others | - | Simhash dimension: 32 | #cluster centers: 16 |
| | - | - | $\eta = 0.1$ |

## A.2 HYPER-PARAMETER SETTING IN ATARI-2600

The hyper-parameter settings in TRPO, TRPO-Hash and CRL$_{TRPO}$ are shown in Table 3. The performance is evaluated over 5 random seeds. The seeds for evaluation are the same for all methods.

Table 3: Hyper-parameter setting of Atari-2600 in Table 1

| | TRPO | TRPO-Hash | CRL$_{TRPO}$ |
|---|---|---|---|
| TRPO batchsize | 100K | | |
| TRPO stepsize | 0.001 | | |
| Discount factor | 0.99 | | |
| Iteration | 500 | | |
| Max length of path | 4500 | | |
| Policy structure | 16 conv filters of size $8 \times 8$, stride 4 | | |
| | 32 conv filters of size $4 \times 4$, stride 2 | | |
| | fully-connect layer with 256 units | | |
| | linear transform and softmax to output action probabilities | | |
| Input pre-processing | grayscale; downsampled to $42 \times 42$ | | |
| | each pixel rescaled to $[-1, 1]$; | | |
| | 4 previous frames are concatenated to form the input state | | |
| Bonus coefficient | - | 0.01 | 0.01 |
| Others | - | SimHash dimension: 64 | Number of clusters: 16 |
| | | | $\eta = 0.1$ |

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
