# OpenReview forum: "Clustered Reinforcement Learning"
_ICLR.cc/2020/Conference — Reject_

### Official Review · AnonReviewer2 · 2019-10-07
**Official Blind Review #2**

**Rating:** 3

**Review:**

This paper presents a clear approach to improve the exploration strategy in reinforcement learning, which is named clustered reinforcement learning. The approach tries to push the agent to explore more states with high novelty and quality.  It is done by adding a bonus reward shown in Eq. (3) to the reward function. The author first cluster states into clusters using the k-means algorithm. The bonus reward will return a high value for a state if the corresponding cluster has a high average reward. When the total reward in a cluster is smaller than a certain threshold, the bonus reward will consider the number of states explored. In the experiments, the authors test different models on two MuJoCo tasks and five Atari games. TRPO, TRPO-Hash, VIME are selected as baselines to compare with. Results show that the proposed bonus reward reaches faster convergence and the highest return in both MuJoCo tasks. In those five Atari games, the proposed method achieves the highest or second-highest average returns.

Although the paper is generally easy to follow and the motivations of the equations are clear,  the analysis of the results is missing and thus the paper provides very limited insights on the behavior of the proposed method. As a result, my opinion on this paper leans to a rejection. As ICLR recommends paper length to be 8 pages, the authors can and should use the remaining space to give more details. For example, the authors can show the mean reward and number of states in all clusters to see whether the agent is efficiently exploring different clusters. And as the number of clusters K is an important hyper-parameter, the reader will also be curious about the resultant performance of the method with different K.

Some questions:
1) In Eq. (3), you have two hyper-parameters in the bonus reward, and for MuJoCo and Atari games, you are using different settings for the first coefficient. How do you choose the hyper-parameter settings? Do you perform grid search with another environment or a set of environments to determine the hyper-parameters?

3) In the algorithm, the method has to learn new cluster assignments in each iteration. Does it significantly slow down the training of the agent?

4) In the experiments, the authors compare with other methods on only five Atari games. However, there should be more environments available.  What is the reason for choosing these five games? Unless it is difficult to gather the scores on more environments, I believe the authors shall provide results with more Atari environments.

5) The conclusion (Section 6) is extremely short and it claims that "CRL can outperform other SOTA methods in most cases". However, for the five Atari games, the only game that CRL achieves the best performance among seven methods is Venture, according to Table 1.
Such a claim will confuse readers as it is not in line with the results.

All in all, I believe this paper can be significantly improved if more details and analyses are provided.


**Experience Assessment:**

I do not know much about this area.

**Review Assessment: Checking Correctness Of Derivations And Theory:**

I did not assess the derivations or theory.

**Review Assessment: Checking Correctness Of Experiments:**

I carefully checked the experiments.

**Review Assessment: Thoroughness In Paper Reading:**

I read the paper thoroughly.

---

### Official Review · AnonReviewer3 · 2019-10-22
**Official Blind Review #3**

**Rating:** 6

**Review:**

This paper proposed a new reinforcement learning framework named clustered reinforcement learning. The proposed method employs the clustering method to explore the novelty and quality in the neighboring area. Some suggestions are as below:

1. The method adopts the k-means for clustering. How about other clustering methods, like Spectral Clustering and other recent deep clustering methods? It's expected to give the experiment comparison results with other clustering methods. Is the method senstive to the used clustering method.
2. Usually in the clustering tasks, how to decide the cluster number K is a crucial problem for many applications. And for this method, the clustering is employed in the exploration stage to cluster neighboring areas, how to determine the value of K. Are there any specifical information could be used? Moreover, could the clustering part be jointly trained in the framework? It' required to investigate the influence of K theoretically and experimentally.
3. As the author claimed, the novelty and quality are both important for exploration in RL, the key question is thus how to balance and utilize them carefully in the exploration stage. The contribution of this paper is about the state clustering for exploration in RL which can further reflect the novelty and quality. How to utilize and balance the novelty and quality is still unsolved in this paper.

In summary, I ack that the idea is effective but seems straightforward. It would be better to present some theoretical analysis.

**Experience Assessment:**

I have published one or two papers in this area.

**Review Assessment: Checking Correctness Of Derivations And Theory:**

N/A

**Review Assessment: Checking Correctness Of Experiments:**

I assessed the sensibility of the experiments.

**Review Assessment: Thoroughness In Paper Reading:**

I read the paper at least twice and used my best judgement in assessing the paper.

---

### Official Review · AnonReviewer1 · 2019-10-23
**Official Blind Review #1**

**Rating:** 3

**Review:**

This paper proposed a clustering based algorithm to improve the exploration performance in reinforcement learning. Similar to the count based approaches, the novelty of a new state was computed based on the statistics of the corresponding clusters. This exploration bonus was then combined with the TRPO algorithm to obtain the policy. The experimental results showed some improvement, compare with its competitors.

Although the proposed method is somewhat similar to the earlier hash based approaches, I think it is still interesting by using the clustering, instead of computing the hash code with neural networks. On the other hand, the motivation and explanation of this method are not well presented. I also have some concern regarding the fairness of the comparison in experiments. The English usage could be improved as well. My detailed comments and questions are as follows.
1. The new proposal for the exploration bonus is provided in Equation (3). The denominator there is essentially the count, which is consistent with previous count based approaches (though not with the square root). For the numerator, I am a bit confused about the choice, as if "N" is small, the accumulated "R" could be small as well, which may offset the bonus based on count. I also didn't understand the author's claim that "...it is highly possible that all states in cluster \phi (s) have zero reward", just below Equation (3). Unless the authors provide more details, I am not convinced that the enumerator could be a good choice.
2. Given the proposed bonus, I am wondering how sensitive could it be to the choice of hyperparameters, especially w.r.t \eta. The authors may need to provide more ablation studies on their effect.
3. Another concern is regarding the scalability of the proposed method. Algorithm 1 implies that k-means needs to be conducted in every iteration, which could be very slow. So how about the running time of the proposed method, when compared with baselines?
4. In the experiments, the authors claimed that the code for TRPO-Hash is provided by its authors. However, the scores for TRPO-Hash were much worse than the numbers in the TRPO-Hash paper (see their Table 1). Do you have any explanation?


**Experience Assessment:**

I have published in this field for several years.

**Review Assessment: Checking Correctness Of Derivations And Theory:**

I assessed the sensibility of the derivations and theory.

**Review Assessment: Checking Correctness Of Experiments:**

I carefully checked the experiments.

**Review Assessment: Thoroughness In Paper Reading:**

I read the paper thoroughly.

---

### Decision · Program_Chairs · 2019-12-19

**Decision:**

Reject

**Comment:**

The paper discusses a simple but apparently effective clustering technique to improve exploration. There are no theoretical results, hence the reader relies fully on the experiments to evaluate the method. Unfortunately, an in-dept analysis of the results is missing making it hard to properly evaluate the strength and weaknesses. Furthermore, the authors have not provided any rebuttal to the reviewers' concerns.